# Modeling Chronic Pain Experiences from Online Reports Using the Reddit Reports of Chronic Pain Dataset

**Diogo A. P. Nunes** [1,2,*] **, Joana Ferreira-Gomes** [3,4] **, Fani Neto** [3,4] **and David Martins de Matos** [1,2]

1    Instituto de Engenharia de Sistemas e Computadores—Investigação e Desenvolvimento, 1000-029 Lisbon, Portugal
2    Instituto Superior Técnico, Universidade de Lisboa, 1049-001 Lisbon, Portugal
3    Department of Biomedicine, Experimental Biology Unit, Centre for Medical Research (CIM), Faculty of Medicine, University of Porto, 4200-319 Porto, Portugal
4    i3S—Instituto de Investigação e Inovação em Saúde, University of Porto, 4200-135 Porto, Portugal
*    Correspondence: diogo.p.nunes@inesc-id.pt

**Abstract:** Reported experiences of chronic pain may convey qualities relevant to the exploration of this private and subjective experience. We propose this exploration by means of the Reddit Reports of Chronic Pain (RRCP) dataset. We define and validate the RRCP for a set of subreddits related to chronic pain, identify the main concerns discussed in each subreddit, model each subreddit according to their main concerns, and compare subreddit models. The RRCP dataset comprises 86,537 submissions from 12 subreddits related to chronic pain (each related to one pathological background). Each RRCP subreddit was found to have various main concerns. Some of these concerns are shared between multiple subreddits (e.g., the subreddit *Sciatica* semantically entails the subreddit *backpain* in their various concerns, but not the other way around), whilst some concerns are exclusive to specific subreddits (e.g., *Interstitialcystitis* and *CrohnsDisease*). Our analysis details each of these concerns and their (dis)similarity relations. Although limited by the intrinsic qualities of the Reddit platform, to the best of our knowledge, this is the first research work attempting to model the linguistic expression of various chronic pain-inducing pathologies and comparing these models to identify and quantify the similarities and differences between the corresponding emergent, chronic pain experiences.

**Keywords:** chronic pain; language of pain; natural language processing; social media; chronic pain modeling; subjective experience modeling

## 1. Introduction

Chronic pain is a major health problem [1], with impacts at the individual, social, and economic levels [2]. Language is a key communicator for the task of clinical chronic pain assessment and management [3,4]: a description of the experience often includes valuable information about the bodily distribution of the feeling of pain, temporal patterns of activity, intensity, emotional and psychological impacts, and others, revealing the multidimensionality of this experience [3]. Additionally, the choice of words may reflect the underlying mechanisms of the causal agent(s) [3], if any, which in turn may be used to redirect therapeutic processes. This linguistic expression has been previously studied, such as in the structuring of the Grammar of Pain [5] and the study of its lexical profile, which resulted in the McGill Pain Questionnaire (MPQ) [6], which is widely used to characterize pain from a verbal standpoint in clinical settings [7,8]. However, all these studies relied on manual methods, expensive human evaluation, and limited sample sizes (e.g., the MPQ was originally developed with only 297 participants).

Language has been explored with increasingly more complex Natural Language Processing (NLP) techniques, both due to the development of said techniques and the larger availability of relevant data, usually in thousands of instances or even more. Specifically, regarding health-related applications, various works started to focus on mental health

due to its close relation with language, such as depression diagnosis [9], suicidal ideation detection [10], and the linguistic analysis of multiple and co-occurring mental health conditions [11]. Indeed, some works have focused on computationally exploring language for chronic pain, such as extracting biomedical entities and relations from disease-specific online forums [12], importance analysis of latent topics (as pre-defined by the authors, as opposed to automatically extracted) in online discussions of Inflammatory Bowel Disease [13], qualitative analysis of the concerns of women with Rheumatoid Arthritis, according to textual submissions to Reddit on specific sub-forums related to this disease [14], and topic modeling over the Reddit's sub-forum ChronicPain to analyze common semantic structures of chronic pain online reports, discovering that back pain is, by far, the most mentioned [15].

Reddit is a social media platform structured in sub-forums (called subreddits), each focused on a given, self-moderated topic. Each subreddit is moderated according to its specific rules, topic(s) of discussion, and quality of the moderation. Considering only public subreddits, any user can participate in accordance with their rules. Additionally, Reddit is implicitly anonymous, i.e., users can choose not to disclose their identity without limiting platform use. Reddit's data are made publicly available through the Reddit API, with the Python Reddit API Wrapper (PRAW) (https://github.com/praw-dev/praw (accessed on 19 April 2021)) and the Python Pushshift.io API Wrapper (PSAW) (https://github.com/dmarx/psaw (accessed on 19 April 2021)). Because of these characteristics, increasingly more research studies are based on Reddit data, including health-related applications [16]. Given user anonymity, Reddit's demographics cannot be easily described. According to the platform administrators (Reddit Inc. Audience. (Online) https://web.archive.org/web/20210117184818/ https://www.redditinc.com/advertising/audience (accessed on 14 July 2022)), in 2021, Reddit is at the top-5 most visited sites in the United States, with over 52 million daily users and more than 100k active subreddits. Regarding age groups, 58% of its users are reported to be between 18–34 years old, 27% between 35–44 years old, and 19% are 45 years old or older. Moreover, 57% identify as male and 43% as female. Regarding user physical location, as of May 2022, some websites report that 47.13% of Reddit.com's internet traffic comes from the United States of America (USA) (Clement J. Regional distribution of desktop traffic to Reddit.com as of May 2022 by country. (Online) https://www.statista.com/statistics/325144/reddit-global-active-user-distribution/ (accessed on 14 July 2022)). Although internet traffic is not conclusive regarding user geographical distribution, it points towards a distributional hotspot. We could not find official sources reporting this information.

In this work, we present the Reddit Reports of Chronic Pain (RRCP) dataset, which comprises social media textual descriptions and discussion of chronic pain experiences, on Reddit, from multiple base pathologies (as represented by subreddits explicitly focused on said pathologies), which are known to be commonly accompanied by chronic pain. We used the RRCP to model the language of chronic pain, as used in that corpus. We started by discovering latent topics of the whole corpus, explicitly describing it in that space, which we called the semantic space. Then, observing only the textual entries of any one given subreddit in this semantic space, we approximated their distribution in that space, identifying regions of high density. These regions, enriched by their latent semantics, allowed us to identify the core concerns, or qualities, of what it is like to experience chronic pain, as reported in that subreddit. The set of concerns of any given subreddit, which we call its semantic span, defines the model of that subreddit. Using graph theory, we compared the semantic spans of every subreddit, allowing us to determine the similarities and differences between distinct experiences of chronic pain, as given by their distinct subreddits. We further explored which concerns were shared by all reported experiences of chronic pain and which were exclusive to specific reported experiences. With this, we show that our findings are useful for gaining insights into what it is like to experience chronic pain, as reported in each subreddit in the RRCP.

To the best of our knowledge, this is the first research work attempting to model the linguistic expression of various chronic pain-inducing pathologies (as found on Reddit) and comparing these models to identify and quantify the similarities and differences between the reported chronic pain experiences.

## 2. Materials and Methods

### 2.1. Data Collection

Our aim was to develop a dataset containing Reddit submissions describing or discussing experiences of chronic pain from multiple pathological perspectives. To that end, we manually selected subreddits whose explicit focus was a pathology known to be commonly accompanied by chronic pain, and, for each of these subreddits, retrieved all textual submissions (i.e., with a body of text) ever posted until 2020 (inclusive). For each submission, we retrieved its unique identifier, URL, date of submission (in Coordinated Universal Time (UTC)), author's username, title, body of text, number of comments, and score. We also retrieved the complete comment tree of each submission (capturing the same data fields), although those are not considered for this work. We used the author's username to determine the number of unique authors per subreddit and the distribution of submissions per author. The collection algorithm is publicly available (https://gitlab.hlt.inesc-id.pt/dapn/rrcp-collection-public).

### 2.2. Data Demographics

To obtain a sense of the demographical distribution of the RRCP submission authors, we extracted explicitly stated demographic information, such as age, binary gender, and location, from their public Reddit activity. We did not extract any other author information. To this end, we ran adapted versions of the public Sherlock (https://github.com/orionmelt/sherlock (accessed on 20 June 2022)) algorithm and publicly available code used in a similar work [17], hereby called Glorianna. Sherlock was set to extract author age, binary gender, and location. Glorianna was set to extract age and binary gender. Notably, none of the algorithms extracts non-binary gender statements due to the lack of NLP techniques and data representation [17]. We used both tools because they complement each other in pattern-matching rules. Finally, to standardize the location data that were extracted, we used the GeonamesCache (https://pypi.org/project/geonamescache/ (accessed on 20 June 2022)) package to convert city and country names to continent names.

### 2.3. Data Preprocessing

The following preprocessing was applied to each RRCP submission: (1) removal of URL, numbers, references to other subreddits or Reddit users, HTML tags, punctuation, multiple white spaces, words with less than 3 characters, and stop words, using SpaCy [18], (3) lower-casing, and (4) tokenization into unigrams and bigrams as defined by Gensim [19]. Regarding submission text length, we set 30 as the minimum number of tokens and performed outlier removal using the Interquartile Range method [20]. We chose this number because it provided a relatively large word co-occurrence window at the document level, and it was large enough that single-sentence submissions were filtered out. We call a preprocessed submission a document.

### 2.4. Subreddit Core Concerns: Semantic Span Similarities

As previously discussed, there are multiple aspects to experiencing chronic pain, such as bodily pain distribution, variations of intensity, difficulties with work life, social life, and so on. These examples are somewhat common to different sources of chronic pain. However, other concerns or qualities may be more specific to certain types of chronic pain experiences. For example, because Crohn's Disease has an important manifestation around the gastrointestinal tract and its functions [21], it is conceivable that concerns about diet are a relevant quality to this specific type of experience, which may not be the case for other emergent experiences of chronic pain, such as chronic migraines. Thus, the objectives of this experiment were three-fold: (1) to identify the concerns, or qualities, of each subreddit, in the semantic space; (2) to determine which concerns are shared among various subreddits and which are specific to only a few; and (3) to attribute meaning to the discovered concerns.

To this end, the RRCP was projected onto a topic space as given by the Latent Dirichlet Allocation (LDA) topic model [22], with $k = 20$ topics, which was empirically determined to

identify regions of interest. We call this *k*-dimensional topic space the semantic space. LDA, and topic modeling, in general, are described in greater detail in Appendix A in the context of the discussion of a baseline analysis of subreddit similarity. The experiment described here is based on the same *k*-dimensional semantic space.

The concerns of a subreddit were given by the intrinsic clusters of the document distribution of that subreddit in the semantic space, i.e., the regions of high density. The clustering algorithm K-Means [23] was used for this. The number of clusters is dependent on the distribution of the documents of each subreddit and was given according to 3 clustering metrics, specifically, the squared sum distance to the closest centroid (i.e., inertia), Calinski-score [24], and the silhouette score [25]. Thus, a subreddit was characterized by a matrix of cluster centroids of dimensions $c_i \times k$, where $c_i$ is the number of clusters of the *i*th subreddit. This matrix defines the subreddit's semantic span. Every pair of cluster centroids of all subreddits was compared in terms of cosine similarity (Equation (A1)).

We assessed the results using a similarity graph. The semantic span of a given subreddit is constructed as a similarity graph by having nodes represent the subreddit centroids and edges represent a cosine similarity $\geq 0.9$ between any two centroids (i.e., nodes). This similarity threshold value was empirically determined to show regions of interest. The semantic spans of multiple subreddits are constructed into a single similarity graph using the same method. We identified the (dis)connected components, or sub-graphs, which, in this context, represent concerns or qualities shared among all subreddits (cliques) and those that are not (e.g., disconnected nodes).

Finally, we associated explicit semantics with each of the discovered sub-graphs by observing the top 10 words of all documents belonging to that sub-graph. Words were ranked according to their Term Frequency/Inverse Document Frequency (TFIDF) score in that subset of documents, which is commonly used in the literature, including health applications [26,27].

## 3. Results

### 3.1. Data Description

The preprocessed RRCP is composed of 86,537 Reddit submissions to a total of 12 subreddits from 2013 to 2020 inclusive. These submissions were posted by 44,815 authors. The subreddit names and total number of submissions per subreddit are shown in Table 1, which also shows the summary of the dataset regarding the number of submissions and tokens per subreddit. The RRCP dataset is described and explored in greater detail in Appendix B. Notice that, in this work, we reference each subreddit by its public name, even if it contains morphosyntactic errors. These references are always italicized in the main body of the text.

**Table 1.** Summary of the RRCP dataset regarding textual data per subreddit. Standard deviation is shown in parentheses. Notice that the number of registered users in each subreddit does not have to necessarily match the number of submission authors of that subreddit.

| Subreddit | Mean Submissions Per Year | Total Number of Submissions | Mean Tokens Per Submission | Total Tokens | Number of Registered Users (Thousands) |
|---|---|---|---|---|---|
| CrohnsDisease | 2854.5 (1569.7) | 22,836 | 127.6 (85.9) | 2,913,623 | 35 |
| migraine | 2363.9 (2238.4) | 18,911 | 134.4 (88.5) | 2,542,288 | 73.9 |
| ChronicPain | 1545.6 (989.6) | 12,365 | 159.7 (99.7) | 1,974,368 | 54.1 |
| fibromyalgia | 1345.5 (1220.4) | 10,764 | 138.6 (90.6) | 1,492,326 | 34.1 |
| lupus | 537.5 (589.2) | 4300 | 135.0 (89.2) | 580,571 | 12 |
| Interstitialcystitis | 437.5 (535.0) | 3500 | 144.5 (96.2) | 505,836 | 9.2 |
| rheumatoid | 407.2 (353.8) | 3258 | 131.7 (84.0) | 429,151 | 12.5 |
| backpain | 333.4 (433.4) | 2667 | 148.0 (92.1) | 394,617 | 15.6 |
| Sciatica | 319.4 (404.9) | 2555 | 156.3 (95.0) | 399,278 | 10.1 |
| ankylosingspondylitis | 306.2 (373.3) | 2450 | 134.5 (90.2) | 329,443 | 8.4 |
| ChronicIllness | 228.0 (329.7) | 1596 | 164.6 (98.1) | 262,716 | 25.5 |
| Thritis | 166.9 (114.4) | 1335 | 142.3 (88.8) | 189,979 | 7.8 |

### 3.2. Data Demographics

Binary gender information was collected for 10,546 (23.53%) RRCP authors. According to these results, 5854 (55.51%) authors identified as being female, and 4692 (44.49%) identified as being male. Table 2 shows the distribution of binary gender per subreddit.

**Table 2.** Binary gender distribution per subreddit.

| Subreddit | Female (%) | Male (%) | Authors with Stated Gender (%) |
|---|---|---|---|
| CrohnsDisease | 40.85 | 59.15 | 20.92 |
| migraine | 63.34 | 36.66 | 24.25 |
| ChronicPain | 54.78 | 45.22 | 28.48 |
| fibromyalgia | 68.24 | 31.76 | 24.71 |
| lupus | 69.35 | 30.65 | 19.43 |
| Interstitialcystitis | 73.26 | 26.74 | 21.65 |
| rheumatoid | 58.58 | 41.42 | 20.13 |
| backpain | 30.04 | 69.96 | 24.18 |
| Sciatica | 32.75 | 67.25 | 21.95 |
| ankylosingspondylitis | 45.45 | 54.55 | 18.06 |
| ChronicIllness | 73.23 | 26.77 | 24.34 |
| Thritis | 51.70 | 48.30 | 29.05 |

Age information was collected for 3090 (6.90%) RRCP authors. According to these results, 193 (6.25%) are 13–17 years old, 2084 (67.44%) 18–34 years old, 413 (13.37%) 35–44 years old, and 382 (12.36%) are 45 years old or older. Table 3 shows the distribution of age per subreddit.

**Table 3.** Age distribution per subreddit.

| Subreddit | 13–17 (%) | 18–34 (%) | 35–44 (%) | >45 (%) | Authors with Stated Age (%) |
|---|---|---|---|---|---|
| CrohnsDisease | 7.17 | 68.93 | 11.95 | 11.95 | 5.78 |
| migraine | 5.59 | 69.57 | 12.67 | 12.17 | 7.49 |
| ChronicPain | 7.16 | 64.41 | 13.37 | 15.07 | 8.27 |
| fibromyalgia | 3.51 | 68.37 | 18.21 | 9.90 | 6.08 |
| lupus | 6.90 | 68.97 | 9.66 | 14.48 | 6.13 |
| Interstitialcystitis | 2.56 | 74.36 | 14.53 | 8.55 | 7.06 |
| rheumatoid | 8.49 | 60.38 | 16.98 | 14.15 | 5.81 |
| backpain | 7.81 | 65.10 | 14.58 | 12.50 | 8.66 |
| Sciatica | 8.99 | 67.42 | 12.36 | 11.24 | 5.66 |
| ankylosingspondylitis | 5.97 | 70.15 | 13.43 | 10.45 | 5.00 |
| ChronicIllness | 3.80 | 77.22 | 12.66 | 6.33 | 7.15 |
| Thritis | 9.52 | 58.33 | 13.10 | 19.05 | 8.30 |

Location information was collected for 9107 (20.32%) RRCP authors. According to these results, 6492 (71.29%) authors reside in North America, 1072 (11.77%) in Europe, 568 (6.24%) in Asia, 466 (5.12%) in Oceania, 300 (3.29%) in Africa, and 209 (2.29%) in South America. Table 4 shows the distribution of location per subreddit.

**Table 4.** Location distribution per subreddit.

| Subreddit | North America (%) | Europe (%) | Asia (%) | Oceania (%) | Africa (%) | South America (%) | Authors with Stated Location (%) |
|---|---|---|---|---|---|---|---|
| CrohnsDisease | 71.39 | 13.30 | 5.76 | 4.84 | 2.99 | 1.73 | 18.46 |
| migraine | 70.24 | 13.11 | 6.69 | 4.33 | 3.23 | 2.40 | 21.30 |
| ChronicPain | 72.71 | 9.56 | 5.11 | 6.57 | 3.78 | 2.26 | 23.45 |
| fibromyalgia | 69.61 | 11.15 | 5.48 | 7.16 | 3.62 | 2.97 | 20.90 |
| lupus | 75.93 | 9.85 | 5.69 | 3.28 | 2.19 | 3.06 | 19.31 |
| Interstitialcystitis | 68.36 | 11.64 | 8.73 | 4.00 | 5.09 | 2.18 | 16.59 |
| rheumatoid | 74.71 | 12.50 | 3.78 | 3.20 | 2.33 | 3.49 | 18.87 |
| backpain | 71.14 | 12.08 | 7.61 | 4.25 | 3.13 | 1.79 | 20.16 |
| Sciatica | 74.68 | 8.44 | 8.44 | 2.92 | 2.60 | 2.92 | 19.59 |
| ankylosingspondylitis | 68.35 | 12.24 | 11.39 | 5.91 | 1.69 | 0.42 | 17.69 |
| ChronicIllness | 67.88 | 11.92 | 7.77 | 4.15 | 6.22 | 2.07 | 17.47 |
| Thritis | 70.29 | 10.46 | 5.86 | 8.37 | 3.35 | 1.67 | 23.62 |

### 3.3. Subreddit Core Concerns: Semantic Span Similarities

The following number of clusters was found for each subreddit: *CrohnsDisease*—7; *migraine*—6; *ChronicPain*—10; *fibromyalgia*—10; *lupus*—12; *Interstitialcystitis*—8; *rheumatoid*—10; *backpain*—5; *Sciatica*—7; *ankylosingspondylitis*—8; *ChronicIllness*—6; *Thritis*—9. These numbers were determined by the clustering metrics defined in the experimental setup. Appendix C shows the results of the clustering metrics for each subreddit.

The sequence Figure 1a–c shows the sequential overlap of the semantic spans of 3 subreddits (*Sciatica, backpain,* and *CrohnsDisease*) to illustrate this experiment's captured core semantics of each subreddit and how they are related between themselves and those of other subreddits. In Figure 1a, we observe the semantic span (i.e., centroids) of *Sciatica*. We also call these the qualities or concerns of that subreddit. According to the applied threshold (cosine similarity $\geq$ 0.9, empirically determined to show regions of interest) in the presented similarity graph, these are all distinct concerns (i.e., nodes are all disconnected). In Figure 1b, we observe the semantic spans of *Sciatica* and *backpain*. According to the similarity graph edges, all *backpain* concerns have a match with one of *Sciatica* (duplets), but not all *Sciatica* concerns have a match with one of *backpain* (disconnected nodes). Finally, in Figure 1c, we observe the semantic spans of *Sciatica, backpain,* and *CrohnsDisease*. According to the similarity graph edges, we observe that *CrohnsDisease* has concerns that match with both *Sciatica* and *backpain*, concerns that match only with *Sciatica* or *backpain*, and concerns that have no match. Notice that, at this stage, we have not associated any meaning with the captured concerns, only their relations.

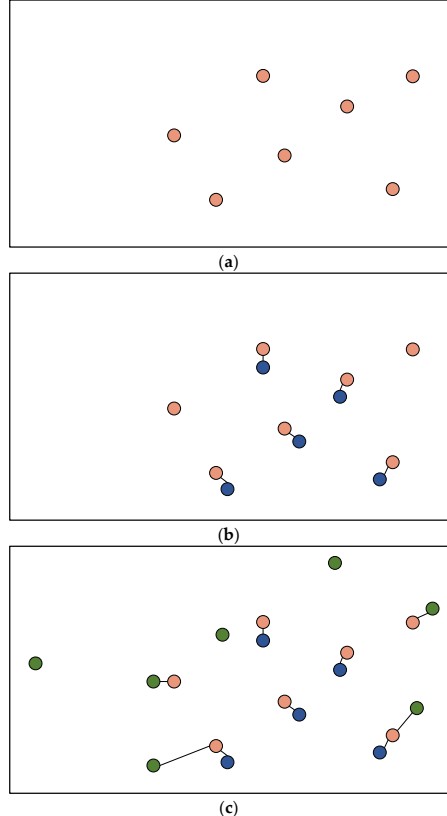

(a)

(b)

(c)

**Figure 1.** (**a**). Semantic span of the *Sciatica* subreddit (light brown). The semantic span of *Sciatica* encompasses 7 centroids. Edges between two nodes represent similarities $\geq$0.9. (**b**). Semantic span of the *Sciatica* and *backpain* (blue) subreddits. The semantic span of *backpain* encompasses 5 centroids. Edges between two nodes represent similarities $\geq$0.9. (**c**). Semantic span of the *Sciatica, backpain,* and *CrohnsDisease* (green) subreddits. The semantic span of *CrohnsDisease* encompasses 7 centroids. Edges between two nodes represent similarities $\geq$0.9.

Figure 2 shows the similarity graph of the semantic spans of all subreddits in the form of a petal graph. In this petal graph, each sub-graph represents one connected component of the similarity graph. Moreover, each of these sub-graphs is characterized by the top-10 TFIDF scoring tokens of the subset of documents that belong to it. Similar to the previous sequential overlap of semantic spans, we observe that certain sub-graphs encompass at least a node from all subreddits (suggesting that that sub-graph is a concern common to all subreddits), others encompass only a subset of subreddits (suggesting that those subreddits have that concern in common, but other subreddits do not), and others are composed of a single subreddit (suggesting that that concern is exclusive to that subreddit, a point of acute dissimilarity).

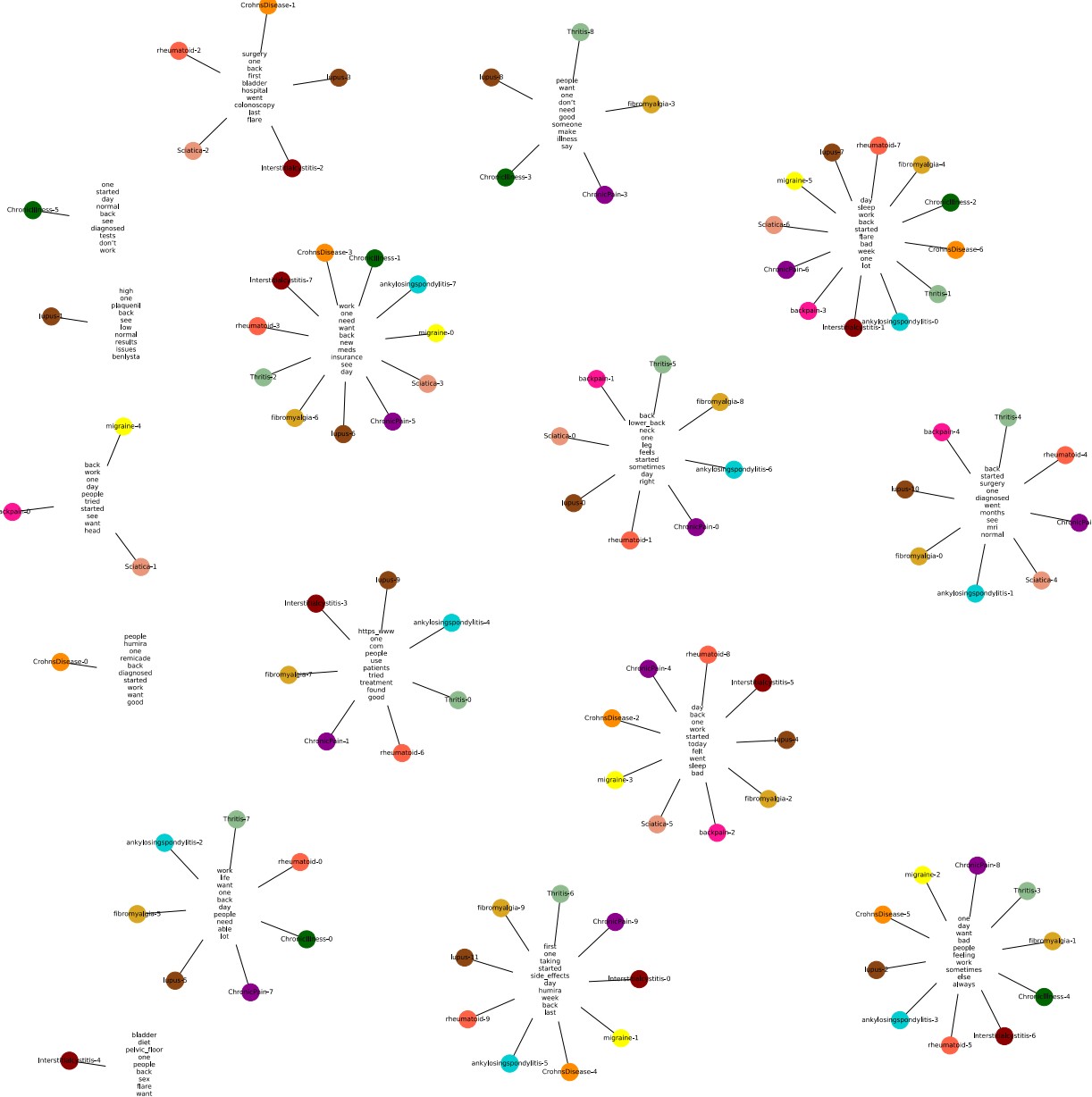

**Figure 2.** Petal graph between all subreddit centroids (nodes, colored by subreddit). Node label indicates the corresponding subreddit and centroid sequential number. Sub-graphs indicate connected components with similarities ≥ 0.9. Each sub-graph is characterized by the top-10 TFIDF scoring tokens of the subset of documents that belong to it.

## 4. Discussion

### 4.1. Data Demographics

A 2022 study [28] on the demographical distribution of chronic pain in the USA revealed that there is a higher likelihood of the presence of chronic pain in people that identify as females of increased age, decreased educational level, and nonmarried status. We extracted demographic information from RRCP authors to understand if this dataset is representative of that populational segment. Because our analysis relied solely on explicit, public statements, not many authors could be characterized, and some only for a subset of the demographic features. Our analysis does not allow for the discussion of the non-binary gender, educational level, and marital status distributions of the RRCP population.

Of RRCP authors, 23.53% were characterized in terms of binary gender, revealing that more than half of the population identifies as female. The majority of subreddits follow the same trend, although some (e.g., *backpain* and *Sciatica*) do not. Importantly, this is opposite to what was observed in the overall Reddit population[3] but aligned with chronic pain's prevalence in female-identified subjects. Although we cannot conclude that the number of reported binary gender data is sufficient to support the small difference in reported female- and male-identified authors, indeed, it is supported by other Reddit-based health studies that observed similar distributions [17].

Only 6.90% of RRCP authors were characterized regarding age. The results are in accordance with the site-wide Reddit reports [3], i.e., leaning towards a younger population but opposed to the reported prevalence of chronic pain in the USA. A similar trend can be observed for each subreddit. The fact that Reddit is an exclusively online social platform limits its use to those with access to and knowledge of such technologies. With this, we cannot conclude that the RRCP population is representative of the overall chronic pain population regarding age distribution.

Regarding author location, we were able to characterize 20.32% of RRCP authors. Most authors have reported to be located in North America. The second most frequent location is Europe, with almost 60 percentage points of difference. Although there are no official reports regarding location distribution of the site-wide Reddit population, it does align with the observed large internet traffic coming from the USA (Clement J. Regional distribution of desktop traffic to Reddit.com as of May 2022 by country. (Online) https://www.statista.com/statistics/325144/reddit-global-active-user-distribution (accessed on 14 July 2022).). Nonetheless, we can conclude that the majority of the RRCP population is comfortable with the English language, which is also the language used in the selected subreddits.

### 4.2. Subreddit Core Concerns: Semantic Span Similarities

We characterized each subreddit by a set of cluster centroids, each representing some semantics discussed by a large volume of documents projected in the semantic space. We called this set of centroids the subreddit semantic span, which is, essentially, a more detailed version of the subreddit centroid explored in the baseline analysis of subreddit similarity (Appendix A). Exactly to overcome the limitations of that baseline experiment, we compared the semantic spans of the various subreddits with similarity and petal graphs. We used the subreddits *Sciatica, backpain,* and *CrohnsDisease* as specific examples of the knowledge this experiment allowed us to extract. The results suggest that the experience of chronic pain as reported in *Sciatica* encompasses the same concerns as the experience of chronic pain as reported in *backpain* and additional concerns which are not relevant in *backpain*. Moreover, that *Sciatica* semantically encompasses *backpain*, and not the other way around. In a similar analysis, the results suggest that *CrohnsDisease* does not fully semantically encompass *Sciatica* or *backpain,* although various concerns are shared. These considerations are aligned with the observations in the baseline experiment (Appendix A). We also observed the similarity graph between the semantic spans of all subreddits in the form of a petal graph, where the high cosine-similarity requisite suggests that each sub-graph represents one concern, shared by all subreddits belonging to it. We observed three types of sub-graphs: cliques (i.e., sub-graphs containing at least one node from all

subreddits), disconnected nodes, and anything in-between. This experimental setup tells us that cliques represent core concerns that are shared between all reported experiences of chronic pain. Sub-graphs that are not cliques and are not disconnected nodes represent concerns that are only relevant to a subset of reported experiences of chronic pain when not all subreddits are represented. Finally, disconnected nodes represent concerns that are exclusive to a specific type of experience of chronic pain, as reported in the corresponding subreddit. Each sub-graph was characterized by the top-10 TFIDF scoring tokens of the subset of documents belonging to it. Even though this is a preliminary, limited approach to attribute meaning to the concepts being discussed by thousands of documents, it was already possible to discern relevant semantics. Accordingly, cliques appear to be common to any experience of chronic pain, e.g., work, feeling sick (or being sick of), doctor, and sleep, and disconnected nodes represent concerns exclusively relevant to one subreddit. For example, the sub-graph on the bottom-left of Figure 2 shows a disconnected node of *Interstitialcystitis*, which appears to be concerned with the bladder, diet, pelvic floor, and others.

Importantly, with this experiment, we observed what the baseline experiment (Appendix A) failed to show: (1) that there are multiple concerns about a single subreddit, as shown by the various nodes of the same subreddit spread out on the semantic space, (2) that there are concerns of reported experiences of chronic pain shared between different subreddits, as shown by connected nodes, and (3) that there are concerns which are exclusive to certain subreddits, as shown by disconnected nodes. These results are in accordance with the known multidimensionality of the experience of chronic pain beyond pain intensity.

### 4.3. Limitations

The presented RRCP dataset is exclusively composed of Reddit content. Regarding Reddit's user base, since it is an online social platform, it is more easily accessible to specific populational segments. Thus, even though we never make generic claims, our conclusions are only applicable to the reported experiences of chronic pain of those segments in this specific social platform. Moreover, although our demographic analysis hints at populational segments partially aligned with the overall chronic pain incidence (e.g., in binary gender distribution), it does not provide sufficient data to state so conclusively. Regarding content, there are also limitations intrinsic to the platform itself. The fact that Reddit is organized in subreddits promotes the development of subreddit-specific cultures, especially in highly active and well-established subreddits, e.g., specific structuring of phrases, use of specific words, and focus on specific topics. Naturally, these all affect the language employed and the topics discussed in any one subreddit, possibly biasing the results of our work. The same applies in the case of subreddit moderation and rules: depending on who and how active the moderators are, the content of a given subreddit might be more or less relevant to our work and more or less limited by subreddit rules. Finally, we have no way to determine if user statements are true or even if they truly experience the pathology topic of the subreddit in which they posted their submissions. The experimental setup described in this work does not account for these limitations.

### 5. Conclusions

In this work, we presented the RRCP dataset, which comprises 86,537 Reddit submissions from 12 subreddits either related to chronic pain directly or to a pathology that is known to be accompanied by chronic pain. We presented an experiment that attempted to reveal the underlying structure and concepts being discussed in the corpus in order to model reported descriptions and discussions of chronic pain on Reddit, and possibly obtain insights about this subjective experience, suggesting underlying semantic structures. It revealed which concepts are discussed and which subreddits are concerned with which concepts. Our approach captured common concepts, such as work life and sleep, to be shared by all subreddits, and other concepts, such as diet and urinary infections, to be exclusive to

specific subreddits. We hope that this work lays the ground for future research by making the RRCP dataset available and validating the semantic analysis with clinical research.

In future work, we point to more intricate approaches to the semantic modeling of the corpus, namely density-based clustering methods. Additionally, the identified sub-graphs were given semantics by the top 10 words of their corresponding documents, which is a baseline approach. Others should be considered, such as multi-document summarization. Moreover, even though the semantic space was defined as the latent topic space, other spaces should be taken into consideration, namely those of pre-trained word embeddings. Finally, the exploration of the presented dataset and the experience of chronic pain as reported on Reddit is not limited to the semantic modeling approach presented in this work. Possibly interesting tasks include symptom extraction from user-generated text, recognition of pain descriptors (for the qualification of pain), and intensity estimation based on keywords (for the quantification of pain). Discarding the possibility of annotating thousands of entries for each of these tasks, all of these must be based on unsupervised methods, which is an interesting challenge.

**Author Contributions:** All authors participated in the experiment design. D.A.P.N. and D.M.d.M. wrote the main manuscript. All authors provided feedback and reviewed the manuscript. All authors have read and agreed to the published version of the manuscript.

**Funding:** Diogo A. P. Nunes is supported by a scholarship granted by Fundação para a Ciência e Tecnologia, with reference 2021.06759.BD. This work was supported by Portuguese national funds through Fundação para a Ciência e Tecnologia, with reference UIDB/50021/2020. This research was supported by the Portuguese Recovery and Resilience Plan (RRP) through project C645008882-00000055 (Responsible.AI).

**Data Availability Statement:** We plan on making the RRCP dataset publicly available upon the acceptance of the paper. Additionally, the data collection algorithm designed for this work is made publicly available at https://gitlab.hlt.inesc-id.pt/dapn/rrcp-collection-public.

**Conflicts of Interest:** The authors declare no conflict of interest.

## Appendix A

In this appendix, we describe a baseline experiment of subreddit similarity using the Reddit Reports of Chronic Pain (RRCP) dataset. We decided to include this experiment in the appendix because it was the motivation for the more intricate experiment presented and discussed in the main body, although the results and conclusions of this experiment are reflected in those of that experiment. Moreover, we use this appendix to describe in greater detail the techniques and technologies used.

### Appendix A.1. Baseline Analysis of Subreddit Similarity

The objective of this experiment was to gain insights into how the various subreddits are related based on what their users described and discussed regarding their experiences of chronic pain. For that, the RRCP corpus was projected onto a latent space as given by the Latent Dirichlet Allocation (LDA) topic model [22], with $k = 20$ topics, which was empirically determined to identify regions of interest. This $k$-dimensional topic space defines the semantic space of the experiments described in this document.

Topic modeling extracts implicit (latent) information in each document belonging to a corpus, explicitly representing them with that information. Thus, each document is projected into the latent space of (abstract) semantic concepts of the corpus, where the value of each dimension represents the weight of that latent topic in the given document. A topic is itself a distribution of weights over the corpus vocabulary, where the weight indicates the level of relevance that word has in the topic, in such a way that the top relevant words of a topic are syntactically and/or semantically related, given that corpus. More specifically, taking a Bag-of-Words (BOW) representation of the corpus (i.e., each document is represented in the corpus vocabulary space, where each dimension is the frequency of a

given word in that document), LDA represents each document as a mixture of multinomial distributions (defined in the latent $k$ topics space), in which each multinomial is defined over the corpus vocabulary space. Each topic mixture $\theta$ is sampled from a $k$-dimensional Dirichlet distribution ($k$ having to be defined a priori), parameterized by $\alpha$, which intuitively models the concentration of topics per document in a collection. Finally, each multinomial is parameterized by another Dirichlet prior $\beta$, which models the concentration of words per topic. This relaxed paradigm allows for many-to-many relationships both between topics and words and documents and topics, which fits the intuition that a document may comprise several topics and that a word may belong to multiple topics. We chose LDA because it is widely used in the literature, including health-related research [12,15,29,30]. We used Gensim's implementation of LDA with default parameters, setting the number of topics to $k = 20$.

In this experiment, each subreddit was characterized by a single point in the semantic space, called the subreddit centroid, allowing for a coarse-grained analysis. The centroid was given by the average of the document-topic vectors of that subreddit. Thus, in this setting, each subreddit was fully characterized by a single vector of length $k$. The subreddit similarity was given by the cosine similarity between the subreddits' centroids, as defined in Equation (A1). This metric ranges from 0 to 1, where 1 is assigned to identical vectors and 0 to orthogonal vectors. Cosine similarity is commonly used to compare vectors because it emphasizes vectors with similar normalized weights assigned to the same dimensions [31–33].

$$\text{sim}(c_i, c_j) = \frac{c_i \cdot c_j}{||c_i|| \times ||c_j||} \tag{A1}$$

We assessed the results using a similarity graph. In this graph, the nodes represent the centroid of each subreddit, and an edge between two nodes represents a cosine similarity $\geq 0.96$, which was empirically determined to show regions of interest. Taking advantage of graph plotting and graph theory, we then identified the (dis)connected components or sub-graphs to draw conclusions.

*Appendix A.2. Results*

Appendix C shows the top 10 words of each of the $k = 20$ topics extracted for the whole corpus. Figure A1 shows the similarity graph between all subreddits. We observe 7 sub-graphs: 4 disconnected nodes, 2 sub-graphs of 2 nodes each, and a sub-graph of 4 nodes, where only one (*rheumatoid*) is connected to all other nodes. Figure A2 presents the same results but in more detail, showing the exact cosine similarity between every pair of subreddits. We observe that, according to our metric, the most similar subreddits are *backpain* and *Sciatica* (0.99), followed by *ankylosingspondylitis* and *Thritis* (0.98). The least similar subreddits are *backpain* and *CrohnsDisease* (0.54), followed by *backpain* and *Interstitialcystitis* (0.56). On average, the subreddit which is most similar to all other subreddits is *lupus* (0.89), and the subreddit which is, on average, least similar to all other subreddits is *backpain* (0.70).

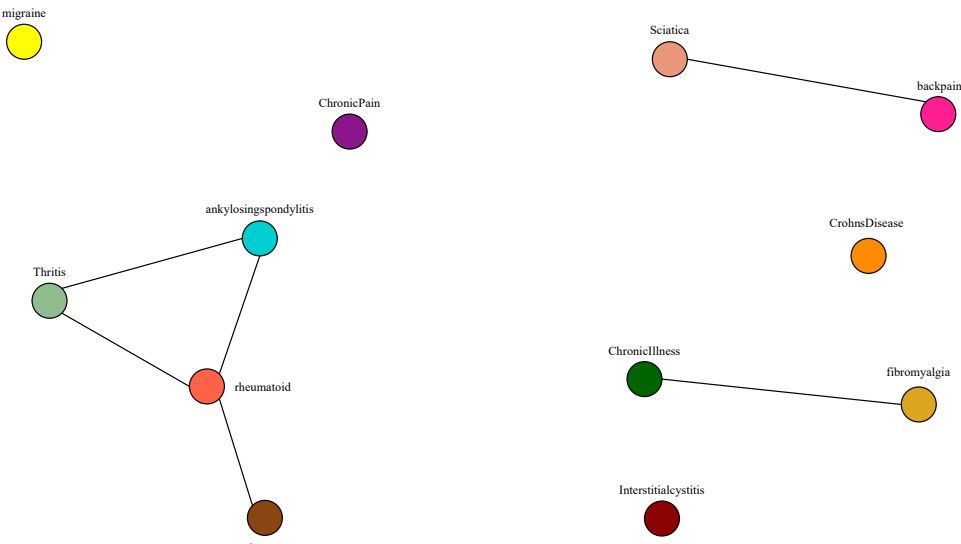

**Figure A1.** Similarity graph between subreddit centroids (nodes). Edges between two nodes represent similarities ≥0.96. Nodes are colored by subreddit. The adjacent label indicates the corresponding subreddit.

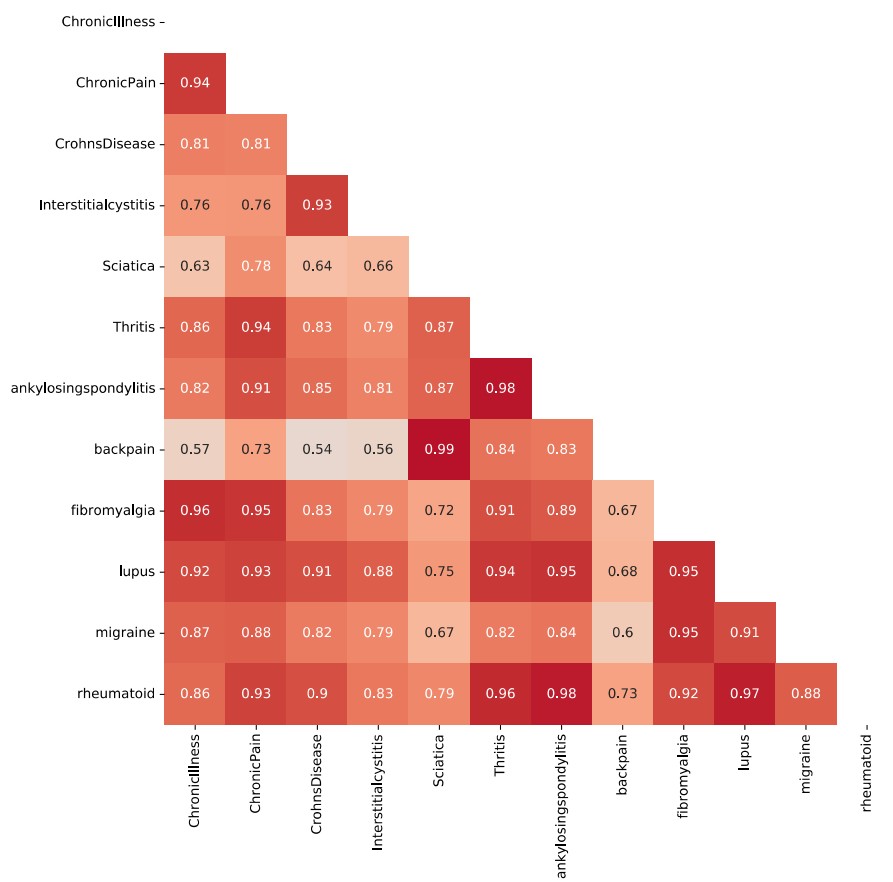

**Figure A2.** Similarity heatmap between subreddit centroids. Annotations represent the similarity between each pair of subreddits.

*Appendix A.3. Discussion*

The results of this experiment reveal important latent structures in the RRCP: overall, the reported experiences of *Sciatica* and *backpain*, *ChronicPain* and *fibromyalgia*, and *Thritis*, *ankylosingspondylitis*, *lupus*, and *rheumatoid* are very similar. Additionally, the remaining reported experiences are considered dissimilar from all the rest. This suggests that, in the RRCP, there are, overall, 7 distinct types of reported chronic pain experiences, although there are 12 distinct subreddits, each focused on a specific pathology. This suggests that there are reports of chronic pain experiences in different subreddits (i.e., of different pathologies) that are, overall, very similar. However, attending to the RRCP characteristics in Table 1 of the main body, we consider it to be sub-optimal to summarize thousands of documents in a single point (i.e., the subreddit centroid) and use it to fully characterize the entire subreddit since this leads to the dilution of meaningful information in the document–topic distribution. Moreover, in determining the similarity between subreddits, there may be specific areas in the semantic space that are very similar between two subreddits and other areas which are very different. This experiment does not capture that information.

**Appendix B**

In this appendix, we describe in greater detail the Reddit Reports of Chronic Pain (RRCP) dataset, which is the one used for analysis in the main body of our work. We analyze the following distributions: (1) subreddit activity, (2) author contribution, (3) submission sentiment, and (4) vocabulary.

*Appendix B.1. Subreddit Activity*

With this analysis, we were interested in learning the user activity distribution of each subreddit and each year, as given by the submission count. This is an important analysis because our main body of work describes experiments that compare subreddits between themselves. These results provide context for those comparisons.

The following metrics were used as measures of user activity, either by subreddit or by year: (1) the number of comments per submission, reflecting the interactivity or discussion stemmed from an average submission, and (2) the length, in tokens, of each submission, which reflects how lengthy the ideas being discussed are in an average submission. Finally, we analyzed the number of submissions per subreddit in a year as an additional measure of user activity.

Starting the analysis by subreddit, we can observe in Figure A3 the comment distribution and in Figure A4 the body length distribution per submission. We conclude that the distribution of comments is similar for all subreddits, rendering comparable the amount of discussion an average submission generates in each of the considered subreddits. Moreover, the distribution of the body length of a submission is very similar for all subreddits, rendering comparable the elaboration of ideas in each of the considered subreddits. According to these metrics, all subreddits have comparable distributions of activity.

Shifting to the analysis by year, we observe the same metrics in Figures A5 and A6, respectively. For both metrics, very similar distributions for each year occur, both presenting a slight tendency to decrease over the years. Nonetheless, according to these metrics, all years have comparable distributions of activity.

Finally, the distribution of the number of submissions in a year per subreddit, as shown in Figure A7, allows us to identify the largest differences between subreddits. According to this metric, *CrohnsDisease* is an outlier, with the following 3 subreddits forming a group and another group formed by the remaining. Excepting *ChronicPain*, which is an all-encompassing topic, it is not clear why the other top-3 subreddits display so much more activity than the other subreddits, especially because they are specifically centered on one pathology. Attending to the number of users of each subreddit and the known population incidence of the considered pathologies, no direct justification was found for these discrepancies.

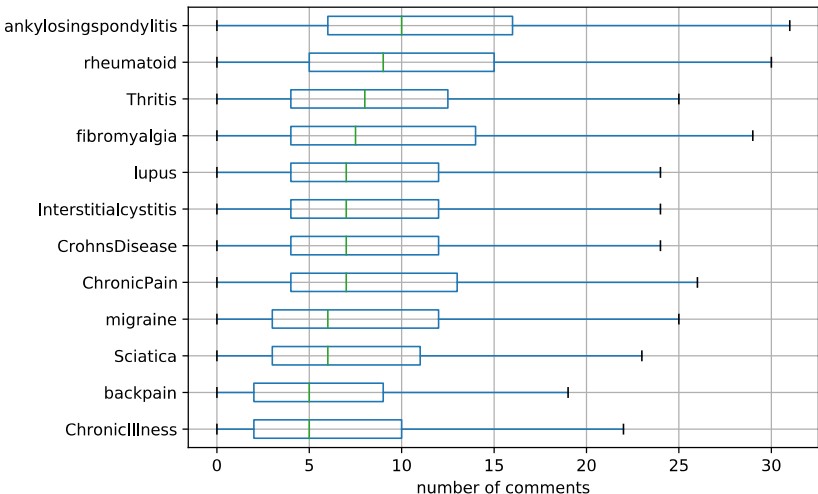

**Figure A3.** Distribution of the number of comments per submission (outliers are omitted).

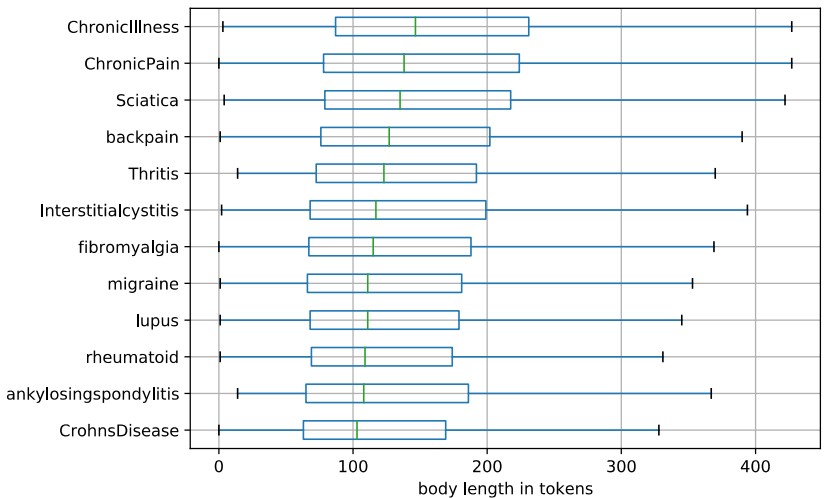

**Figure A4.** Distribution of the body length in tokens per submission (outliers are omitted).

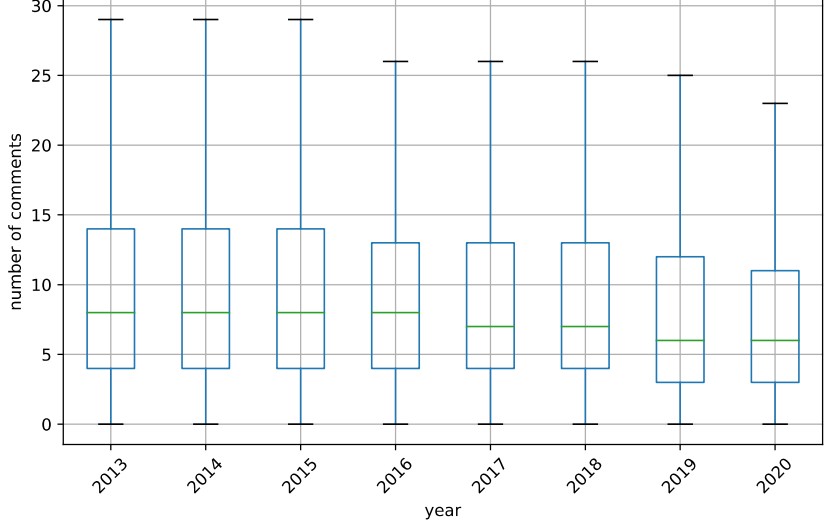

**Figure A5.** Distribution of the number of comments per year (outliers are omitted).

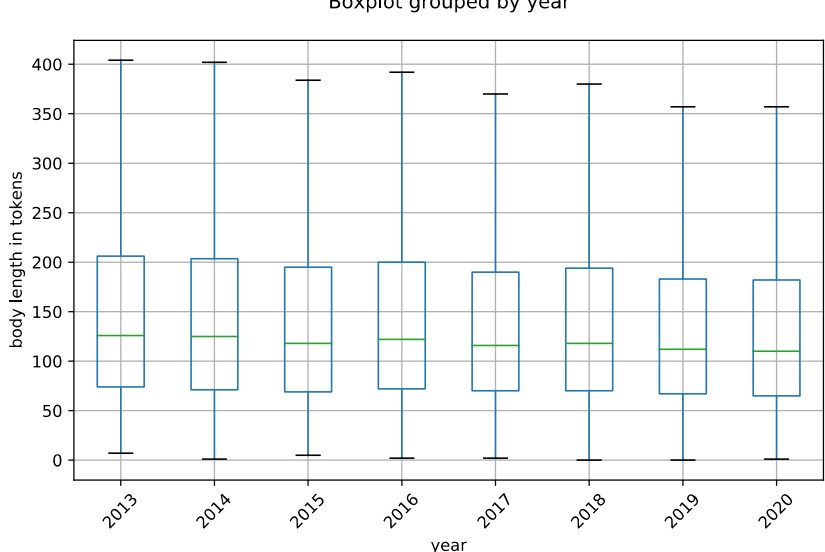

**Figure A6.** Distribution of the body length in tokens per year (outliers are omitted).

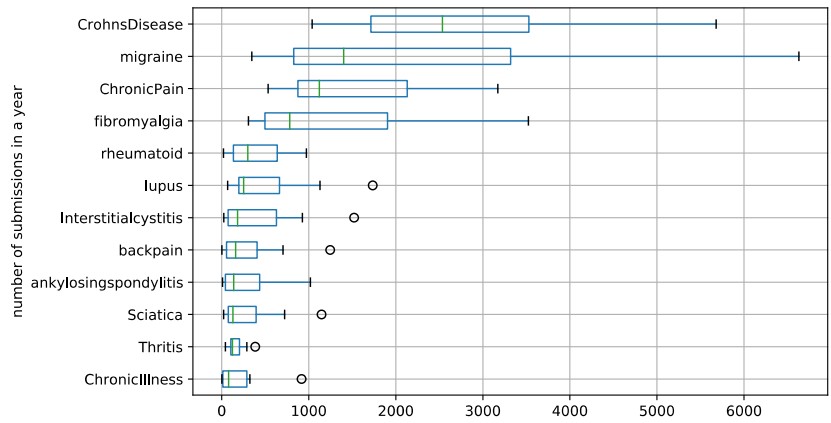

**Figure A7.** Distribution of the number of submissions per year for each subreddit.

*Appendix B.2. Author Contribution*

With this analysis, we were interested in learning the distribution of author contribution, as given by the number of submissions per author. This is an important analysis because it can reveal textual biases due to the over-contribution of specific authors.

The preprocessed RRCP dataset contains 44,815 unique authors. By design, authors only posted submissions to one of the 12 subreddits. Table A1 shows the number of authors and submissions per subreddit. This table shows that the number of authors per subreddit is proportional to the number of submissions per subreddit (Pearson's r = 0.98).

Assessing the contribution of each author in each subreddit, Figure A8 shows the percentage of submissions per author for each subreddit. We can observe that all subreddits have a similar distribution of author contributions. Figure A9 shows the variations between these distributions (percentage of author contribution). We observe that the mean, first, second, and third quartiles of each subreddit's distribution have a variation of less than 0.2 percentage points. Moreover, there is no single author that contributes to a large percentage of one subreddit's submissions (the maximum being 2.2%). These findings point to there not being relevant textual bias due to the over-contribution of a single author or a restricted group of authors for both the whole RRCP and for each subreddit.

**Table A1.** Number of authors and submissions per subreddit. Percentage in relation to the whole dataset is shown in parentheses.

| Subreddit | Authors | Submissions |
|---|---|---|
| migraine | 10,741 (24.0) | 18,911 (21.6) |
| CrohnsDisease | 9409 (21.0) | 22,836 (26.4) |
| ChronicPain | 6423 (14.3) | 12,365 (14.3) |
| fibromyalgia | 5148 (11.5) | 10,764 (12.4) |
| lupus | 2367 (5.3) | 4300 (5.0) |
| backpain | 2217 (4.9) | 2667 (3.1) |
| rheumatoid | 1823 (4.1) | 3258 (3.8) |
| Interstitialcystitis | 1658 (3.7) | 3500 (4.0) |
| Sciatica | 1572 (3.5) | 2555 (3.0) |
| ankylosingspondylitis | 1340 (3.0) | 2450 (2.8) |
| ChronicIllness | 1105 (2.5) | 1596 (1.8) |
| Thritis | 1012 (2.3) | 1335 (1.5) |

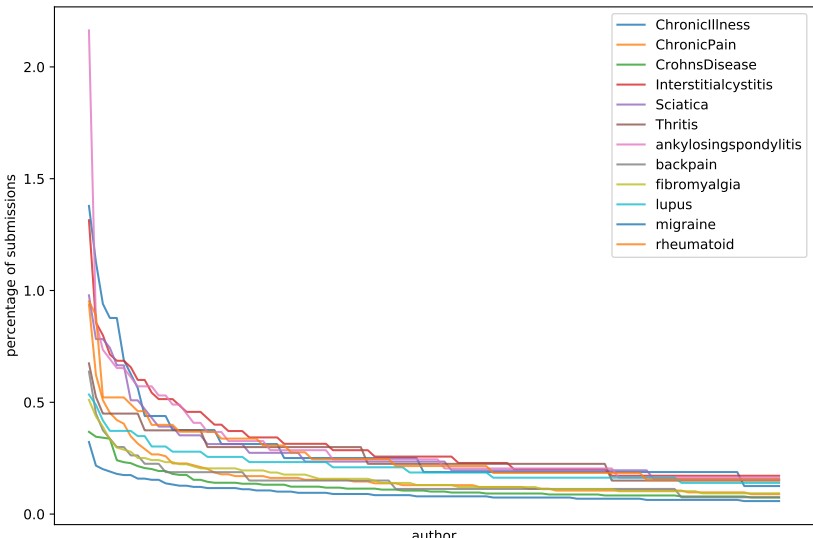

**Figure A8.** Percentage of submissions per author for each subreddit.

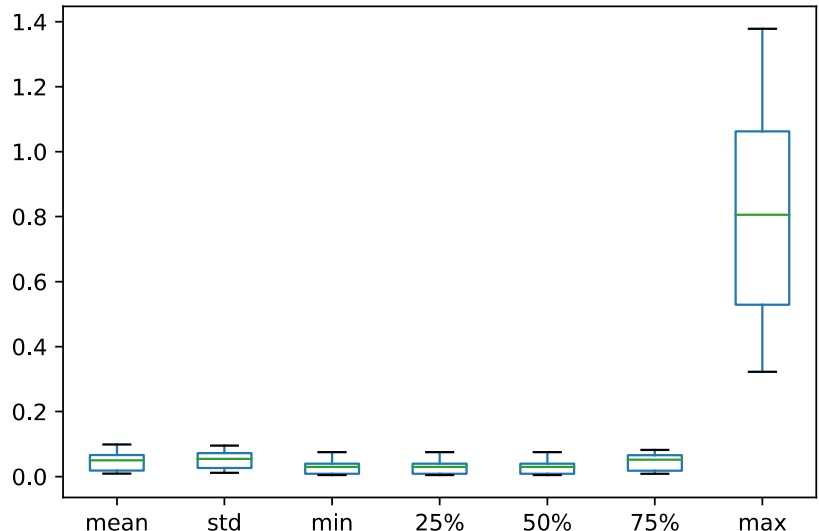

**Figure A9.** Variations between the distribution of percentage of author contribution per subreddit (outliers are omitted).

*Appendix B.3. Submission Sentiment*

With this analysis, we were interested in learning the distribution of sentiment in the RRCP as a whole and per subreddit. Specifically, we were interested in understanding if submissions generally and per subreddit tend more towards a negative, neutral, or positive sentiment. We used the Valence Aware Dictionary and sEntiment Reasoner (VADER) sentiment analysis engine [34], specifically designed for social media text. Even though this engine outputs a continuous sentiment score ranging between −1 (most negative) and 1 (most positive), we used the proposed standardized thresholds to classify a given document as being of positive sentiment (score >= 0.05), neutral sentiment (−0.05 < score > 0.05), and negative sentiment (score <= −0.05). This analysis was motivated by the possibility that the types of forums (i.e., subreddits) explored in our work tend to be more negative and incentivize catastrophizing, losing relevancy for the overall population.

The results of this analysis classify 38,964 (45.03%) documents as negative sentiment, 29,027 (33.54%) as positive sentiment, and 18,546 (21.43%) as neutral sentiment. Figure A10 shows the ratios of negative, neutral, and positive sentiment documents per subreddit in a stacked bar plot. This figure shows that some subreddits, especially *backpain*, *ChronicPain*, and *Sciatica*, tend more toward negative sentiment than the remaining subreddits. All subreddits show a similar ratio of neutral sentiment submissions. Indeed, these results seem to suggest a tendency towards negative sentiment submissions; however, the discrepancy is not large enough to sustain the hypothesis that catastrophizing is the norm. Indeed, it seems natural that reports of chronic pain experiences tend slightly toward negative sentiment.

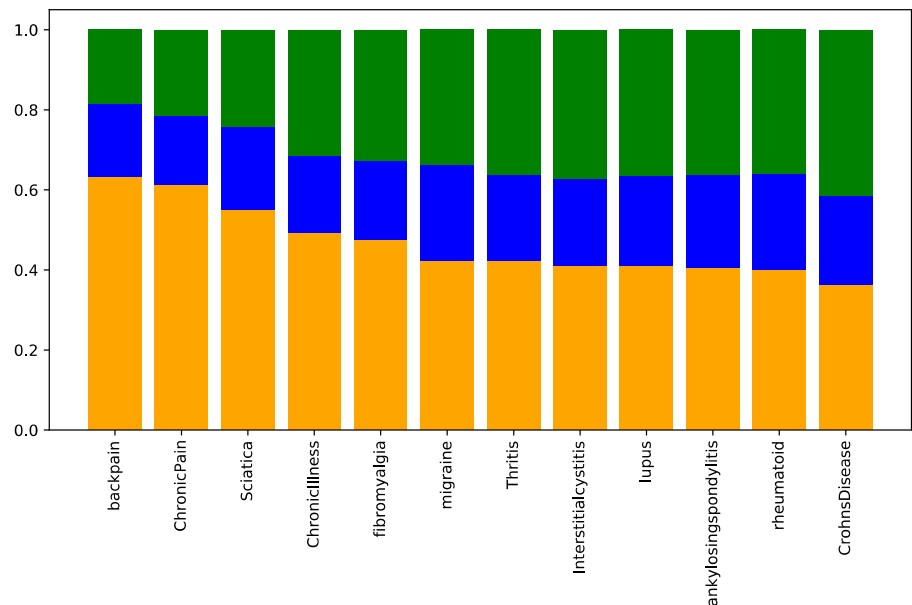

**Figure A10.** Ratio of negative (orange), neutral (blue), and positive (green) sentiment documents per subreddit.

*Appendix B.4. Vocabulary*

The corpus vocabulary is composed of 84,108 unique tokens (unigrams and bigrams). Table A2 shows the top unigrams and bigrams in terms of corpus coverage (i.e., the percentage of documents in the corpus in which they appear). According to this table, all tokens have a corpus coverage below 25%, and all bigrams have a corpus coverage below 5%. The low corpus coverage indicates that there is indeed a very diversified vocabulary. An omitted, additional analysis showed that the least representative tokens mostly corresponded to miscellaneous morphosyntactic errors.

Table A3 shows the top unigrams and bigrams per subreddit, along with the percentage of terms which are exclusive to each subreddit. This table shows that, in total, 48.03% of the vocabulary is not shared between subreddits (i.e., only 55.91% of the corpus vocabulary is shared). This is an important indicator that a large part of the vocabulary used to express, describe, and discuss experiences of chronic pain is exclusive to the type of experience under consideration. It also suggests a significant separation of subreddits in the vocabulary space. We observe that the unigrams back and work are at the top list of almost all subreddits, possibly relating to the location on the body and problems with work life, respectively. It is not clear to us if these unigrams should be considered stop words: on the one hand, they are irrelevant for differentiation between subreddits; on the other hand, they are very relevant concepts surrounding the common experience of chronic pain. We decided to maintain these tokens due to their clinical relevance.

**Table A2.** Top 10 tokens with most corpus coverage, separated as unigrams and bigrams. Percentage is shown in parentheses.

| Unigrams | Bigrams |
|---|---|
| back (21.1) | side effects (4.1) |
| day (18.1) | every day (2.9) |
| work (16.5) | lower back (2.8) |
| bad (14.3) | months ago (2.5) |
| started (14.1) | last week (2.4) |
| first (14.1) | last year (2.4) |
| want (14.0) | last night (2.1) |
| else (13.3) | two weeks (2.1) |
| see (12.8) | came back (1.9) |
| last (12.5) | weeks ago (1.8) |

**Table A3.** Top 3 most frequent unigrams and bigrams per subreddit, along with the percentage of terms which are exclusive to each subreddit.

| Subreddit | Unique Vocabulary (%) | Top-3 Most Frequent Unigrams | Top-3 Most Frequent Bigrams |
|---|---|---|---|
| CrohnsDisease | 11.9 | back, day, first | side effects, last week, last year |
| migraine | 9.6 | day, work, started | side effects, every day, last night |
| ChronicPain | 8.8 | back, day, work | lower back, physical therapy, every day |
| fibromyalgia | 5.6 | work, day, back | brain fog, side effects, every day |
| lupus | 2.4 | back, diagnosed, day | came back, blood work, side effects |
| Interstitialcystitis | 2.0 | bladder, flare, uti | pelvic floor, flare-ups, came back |
| backpain | 1.6 | back, work, day | lower back, upper back, right side |
| rheumatoid | 1.5 | started, day, diagnosed | side effects, months ago, blood work |
| ankylosingspondylitis | 1.4 | back, diagnosed, humira | lower back, ankylosing spondylitis, side effects |
| Sciatica | 1.3 | back, surgery, leg | lower back, herniated disc, left leg |
| ChronicIllness | 1.1 | people, illness, work | chronically ill, brain fog, mental health |
| Thritis | 0.8 | back, bad, work | side effects, months ago, lower back |

## Appendix C

In this appendix, we show the top 10 words of the 20 extracted topics of the whole Reddit Reports of Chronic Pain (RRCP) corpus (Table A4) and the clustering metrics for each subreddit (Figures A11–A22), which allowed for the definition of the number of clusters to extract, per subreddit.

**Table A4.** Top 10 most weighted words of the 20 extracted topics from the Reddit Reports of Chronic Pain (RRCP) dataset.

| Topic Number | Top-10 Most Weighted Words |
|---|---|
| 0 | taking, side effects, tried, medication, meds, started, day, methotrexate, daily, try |
| 1 | don't, high, Topamax, experience, brain fog, wondering, low, see, use, blood pressure |
| 2 | eat, food, eating, drink, water, trigger, day, drinking, triggers, stomach |
| 3 | sensitivity, triggered, bent, tolerate, benlysta, next appointment, long periods, bar, drain, cream |
| 4 | people, life, want, lot, diagnosed, don't, advice, always, way, love |
| 5 | neck, don't, back, cold, won't, idk, caffeine, head, coffee, right side |
| 6 | diet, exercise, weight, diagnosed, lot, trying, months, tried, good, study |
| 7 | head, sometimes, one, else, usually, feeling, always, feels, almost, weird |
| 8 | hair, skin, red, hot, wear, heat, products, use, shower, sun |
| 9 | work, need, appointment, patients, see, office, one, insurance, called, new |
| 10 | diagnosed, test, normal, diagnosis, fatigue, pregnancy, pregnant, stomach, inflammation, crohn's |
| 11 | botox, Enbrel, rheum, glasses, pcp, wouldn't, right eye, ear, ears, hair loss |
| 12 | started, first, last, week, flare, months, weeks, went, back, prednisone |
| 13 | looking, one, find, use, good, people, found, recommendations, helpful, wondering |
| 14 | dizziness, biologics, biologic, wondering, experience, sex, liver, curious, tips, advice |
| 15 | surgery, back, done, recovery, one, mri, long, good, procedure, surgeon |
| 16 | new, month, insurance, treatment, work, one, months, medication, humira, meds |
| 17 | work, day, one, today, bad, want, don't, need, home, sleep |
| 18 | https www, com, https, watch, video, link, articles, gets better, patients, everyone |
| 19 | back, started, joints, hurt, went, joint, lower back, one, day, right |

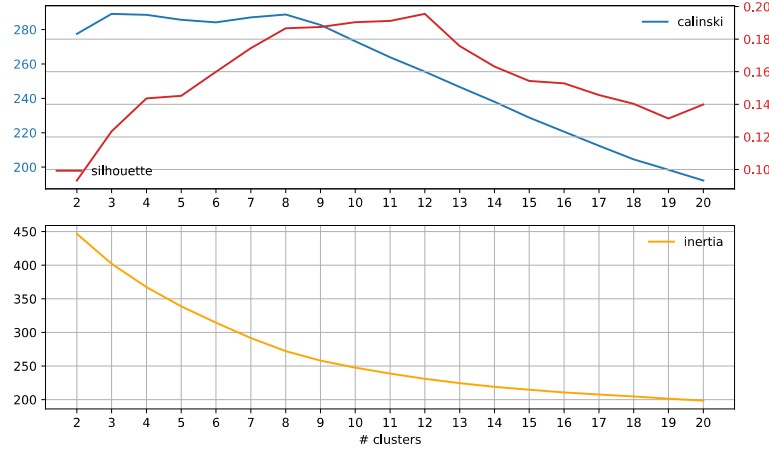

**Figure A11.** Subreddit ankylosingspondylitis.

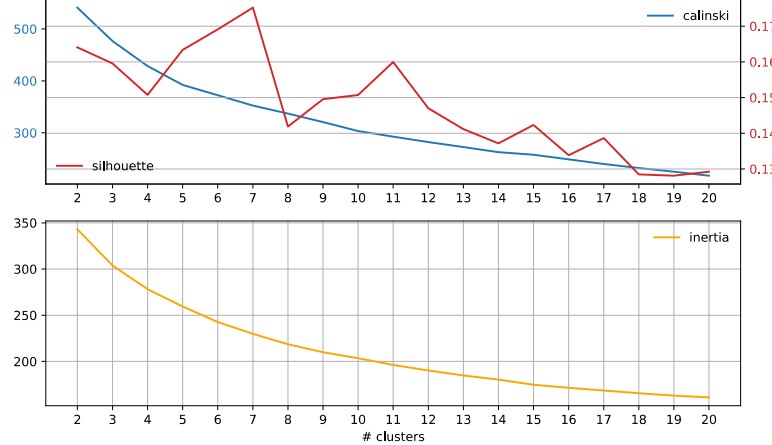

**Figure A12.** Subreddit backpain.

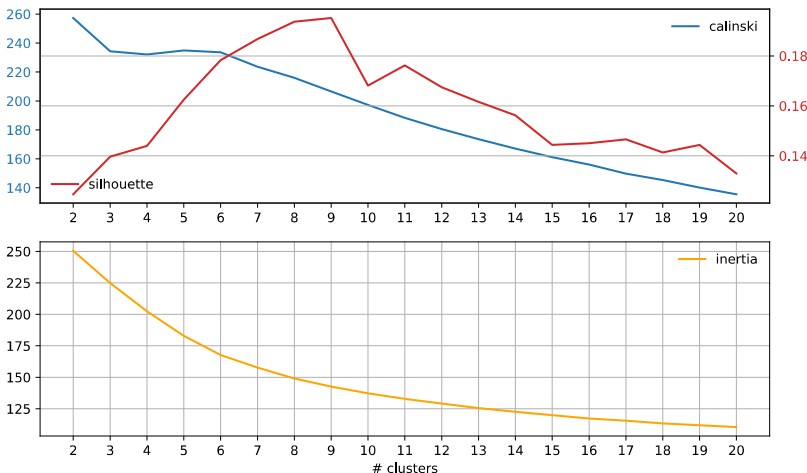

**Figure A13.** Subreddit ChronicIllness.

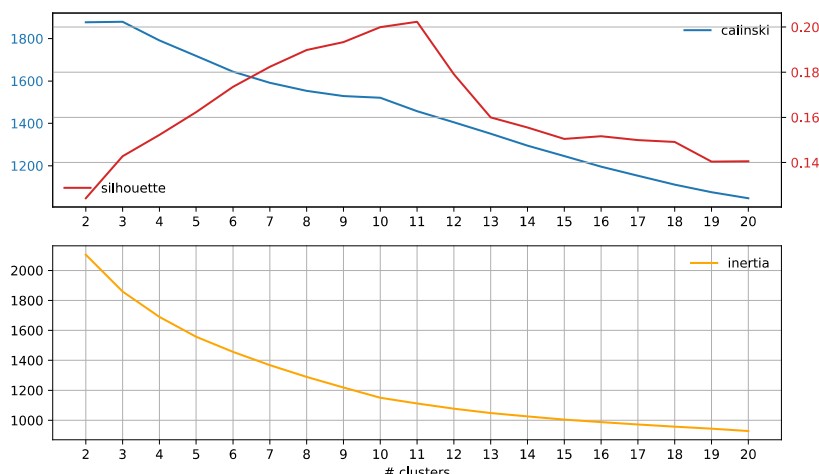

**Figure A14.** Subreddit ChronicPain.

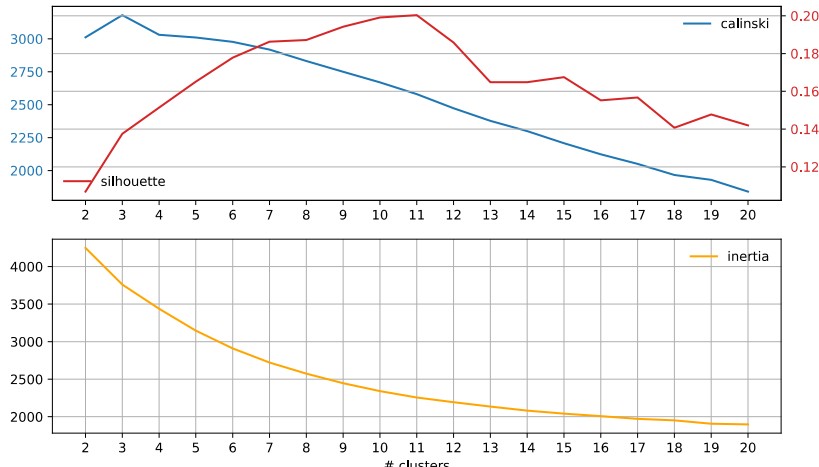

**Figure A15.** Subreddit CrohnsDisease.

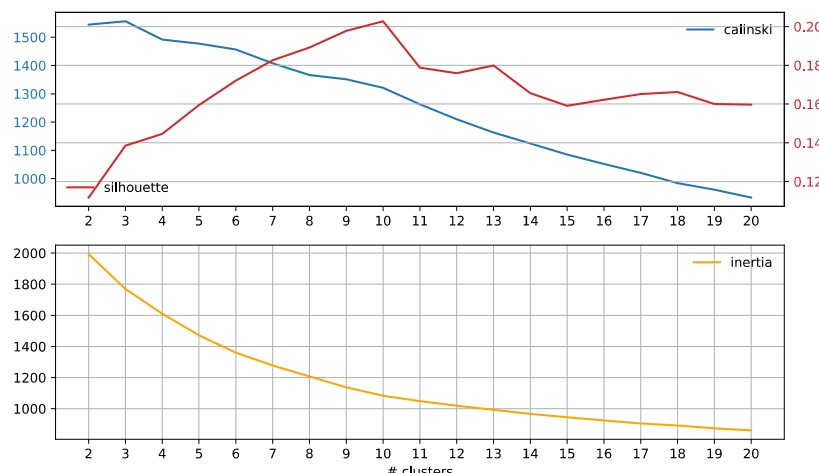

**Figure A16.** Subreddit fibromyalgia.

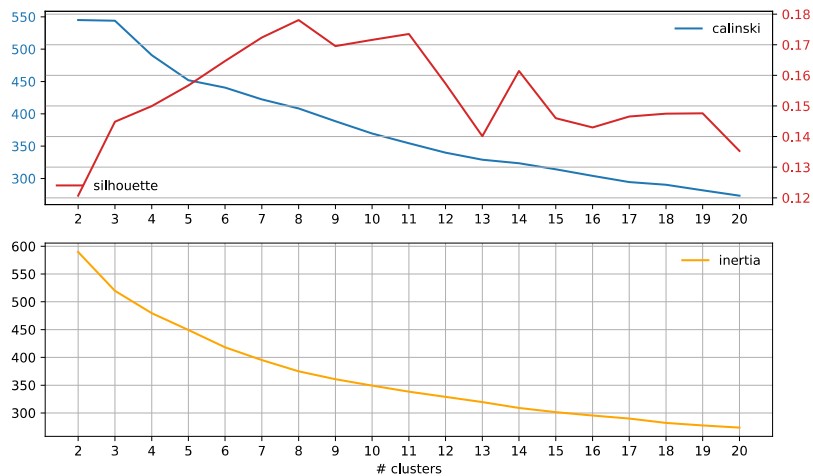

**Figure A17.** Subreddit e.

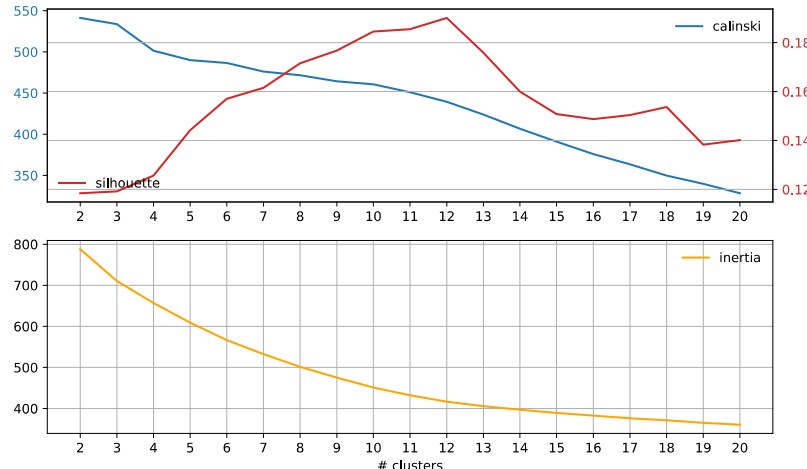

**Figure A18.** Subreddit e.

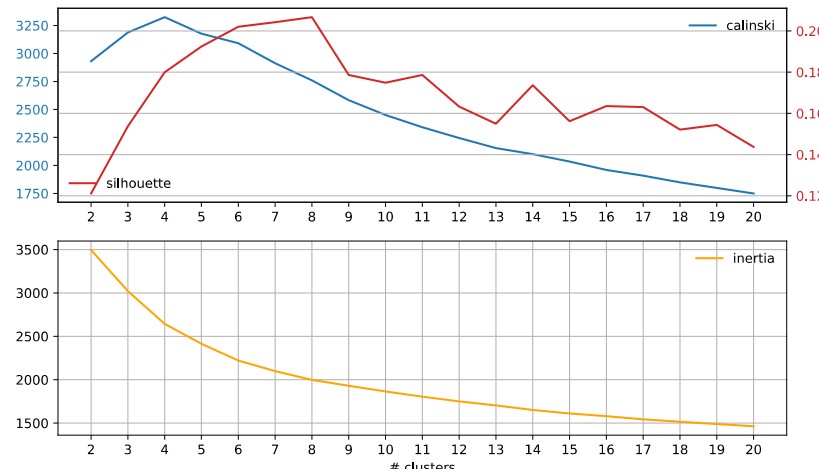

**Figure A19.** Subreddit migraine.

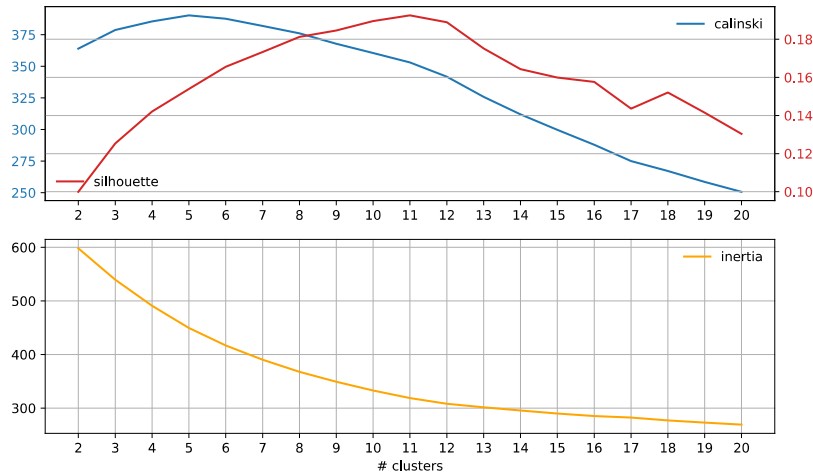

**Figure A20.** Subreddit rheumatoid.

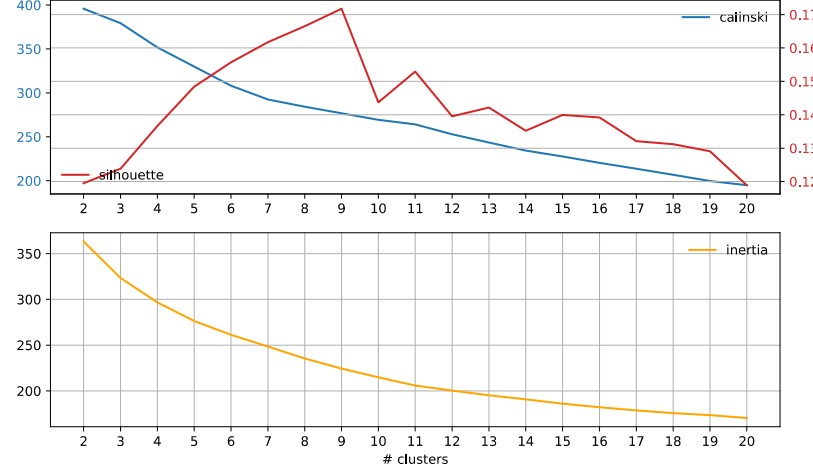

**Figure A21.** Subreddit Sciatica.

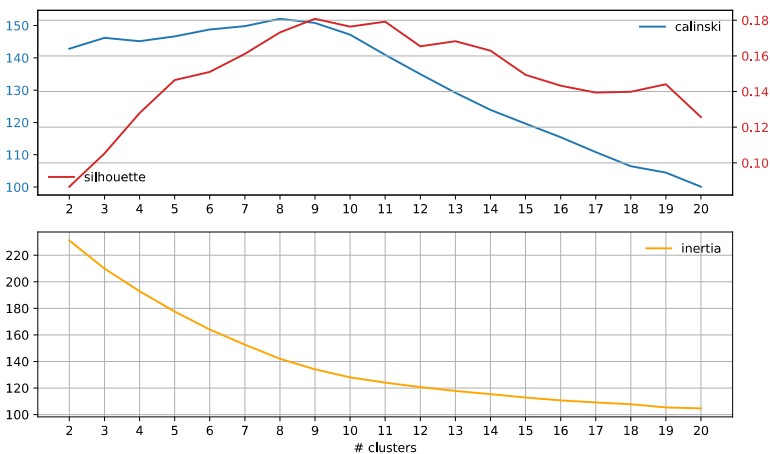

**Figure A22.** Subreddit Thritis.

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
