# Peer review of "Modeling Chronic Pain Experiences from Online Reports Using the Reddit Reports of Chronic Pain Dataset"

_information, doi:10.3390/info14040237_

Round 1

Reviewer 1 Report

The presented study is interesting as the authors explore correlations among chronic pain in various subreddits. The general flow of the paper is really good and easy to follow while also presenting sound reasoning backed up by appendices. There are only some minor concerns to be addressed before considering the manuscript for acceptance:

- at the end of section 1, it can be useful to briefly discuss the differences with citations [12][13][14] with respect to how they analyze chronic pain even though this paper presents a novel dataset (to be published after acceptance)

- figures 1a/b/c might be improved by also showing the concepts associated with the various nodes. Although they are discussed in the appendix, they would allow future readers to understand the images better when they are first introduced.

Reviewer 2 Report

Dear authors, we thank you for considering our journal to present your work on Modeling Chronic Pain Experiences from Online Reports using 2 the Reddit Reports of Chronic Pain dataset.

Although everything can be improved, I believe that the article in its current format can be published because it meets all the requirements for it. The references on which they are based and which are cited in the introduction are correct, as is the research methodology. It provides interesting results and the conclusions respond to the objectives set.